# Quasi-Static Compression Response of Carbon Fiber Reinforced 2.5D Woven Composites at Different Loading Directions

**DOI:** 10.3390/ma15113953

**Published:** 2022-06-01

**Authors:** Tianyu Li, Lifeng Chen, Wei Shen, Lvtao Zhu

**Affiliations:** 1International Institute of Silk, College of Textile Science and Engineering, Zhejiang Sci-Tech University, Hangzhou 310018, China; 202030202116@mails.zstu.edu.cn; 2Shaoxing Baojing Composite Materials Co., Ltd., Shaoxing 312000, China; vincentchenli@sina.com (L.C.); shenw@jinggonggroup.com (W.S.); 3Shaoxing-Keqiao Institute, Zhejiang Sci-Tech University, Shaoxing 312000, China

**Keywords:** 2.5D woven composites, compression performance, X-ray computed tomography, failure mechanism

## Abstract

2.5D woven composites have been increasingly used in aerospace and military applications due to their excellent mechanical properties. In this research, 2.5D woven composites were produced, and their compression responses were investigated in different directions by compression experiments. XR-CT (X-ray computed tomography) technology was used to observe the microstructural damage profiles, and to analyze the failure mechanism of the material. The results show that when subjected to compression loads, the maximum load-bearing capacity of the material in the thickness direction was better than the maximum load-bearing capacity in the warp and weft directions. The compressive strength of the material in the warp and weft directions was lower than that in the thickness direction, and compression damage patterns in each direction also differed.

## 1. Introduction

Three-dimensional composites are a new type of composite material developed by using advanced textile technologies such as weaving, knitting, and splicing [1,2]. Three-dimensional composites reinforced with high-performance carbon fibers have been increasingly used in aerospace and military applications because of their high specific strength, wear resistance, excellent resistance to delamination, and high fracture toughness compared to other structural fiber-reinforced composites [3,4,5].

In the last two decades, a substantial number of studies on composite laminates in various aspects have been conducted, and some of those studies can be found in reference [6,7,8,9,10,11,12,13]. There has been great resistance to their application in many fields, due to the disadvantages of laminates, such as low resistance to interlaminar shear strength and weak impact resistance. To solve this problem, the researchers came up with the concept of three-dimensional composite materials, and a variety of three-dimensional structures of fiber-reinforced composites have emerged due to their good designability. Studies have shown the significant effect of the structure of the prefabricated parts on the performance of carbon fiber reinforced composites [14,15]. The 2.5D woven composite is a special form of 3D woven composite in which the warp yarns and different layers of weft yarns are interwoven at an angle in the thickness direction to form an interlock [16]. Therefore, the 2.5D composites offer better integrity and interlaminar shear resistance than 2D textile composites [17,18] and, at the same time, have lower manufacturing costs and shorter production cycles than 3D composites. As a result, they have attracted increasing attention from scholars both at home and abroad. Duan, et al., analyzed the bending progressive damage behaviors of 2.5D woven SiCf/SiC composites by acoustic emission (AE) technique. The results showed that the damage mechanism of the 2.5D woven SiCf/SiC composite was related to the structure of the preform and the direction of testing, with the material showing pseudo-ductile fracture along the warp direction and brittle fracture along the weft direction [19]. Younes, et al., carried out an optimal design of damage tolerance and modulus of elasticity for a representative volume cell of a 2.5D woven composite based on the three-dimensional Tsai-Wu strength criterion, and validated it against experimental results [20]. Naik, et al., used a stiffness model to predict the elastic and stiffness properties of 3D angle interlock woven composites in uniaxial static tension and shear, and concluded that the overall structure of 3D corner-interlocked woven composites gave the material such advantages as delamination resistance, impact resistance, and dimensional stability [21]. Li, et al., developed a new 3D geometric model to predict the fine mechanical response and effective elastic properties of 3D woven composites by incorporating the morphology, cross-sectional deformation, and spatial orientation of 3D angle-interlocked composite yarns. The results showed that the 3D finite element model accurately simulated the spatial geometric characteristics of the 3D corner-interlocked woven composites. Reasonable overall stress fields and local stress distributions could be identified, which confirmed the strength prediction [22]. SUN, et al., investigated the high strain rate compression behavior of 3D corner-interlocked composites through Hopkinson experiments, and they found that the stress-strain curve was sensitive to the strain rate, and that the maximum stress increased linearly with the strain rate [23]. They proposed that when designing composite structures for 3D woven composites under high strain loads such as shock loading, mechanical parameters with high strain rates should be used. Hallal, et al., presented an improved three-stage homogenization method for 2.5D interlocking woven composites. An analytical model based on a combined hybrid isostrain and isostress model (stiffness and compliance averaging model) was proposed. The results showed that more accurate estimates could be obtained based on a hybrid isostrain and isostress model. The method and the geometric model agreed well with the results of finite element models and relevant experimental data, through which the finite element model was better able to show the potential for the elastic properties of corner-interlocking woven composites [24]. In addition, the researchers also found that defects such as holes, resin-rich regions, and fiber dislocations had a great impact on the mechanical properties of composites. Analyzing the influence of these defects on the failure behavior of materials was of great significance for the safe application of composite materials [25,26,27,28]. Previous researchers have conducted corresponding studies on the mechanical properties of 2.5D woven composites, from simple to complex, from two-dimensional to three-dimensional, and some research results have been obtained, which have laid a solid foundation for subsequent research [29,30,31].

Although domestic and foreign scholars have carried out some explorations of the mechanical properties of 3D woven composites, there were relatively few studies on the mechanical properties in different directions [32,33,34]. The study of loading damage in 3D textile composites therefore still needs to be further developed, and previous researchers have mostly focused on the dynamic mechanical properties of 3D woven composites. In practical engineering applications, 3D textile composites are also subjected to various loads in a quasi-static environment, which can lead to damage. It is therefore particularly important to study their mechanical properties in a quasi-static environment as well.

In this study, 2.5D woven composites were prepared using the VARTM (vacuum-assisted resin transfer molding) technique, and the quasi-static compression response of carbon-fiber-reinforced 2.5D woven composites was investigated experimentally. We aimed to compare the compression response of 2.5D woven composites in different directions, and to analyze the failure mechanism of the material by observing the microscopic damage pattern using XR-CT (X-ray computed tomography), which has rarely been addressed before. Our study provides a beneficial reference for subsequent research on the development of new composite products by studying the compression response of 2.5D woven composites.

## 2. Materials and Methods

### 2.1. Materials

For this study, the detailed weave style of the prefabricated part, consisting of two sets of yarns, was a type of 2.5D woven reinforcement: angle interlock structure between layers. The 2.5D woven prefabricated part consisted of warp and weft yarns, which were oriented along x-direction and y-direction, respectively, (shown in Figure 1b). Here, both warp and weft yarns were 800 Tex. Warp yarns interlocked with different layers of weft yarns through the thickness direction, which resulted in considerably enhanced mechanical properties in the thickness direction of the 2.5D woven composite material. The carbon fiber raw material was supplied by Zhejiang Jingong Carbon Fiber Co., Ltd. (Shaoxing, China), and woven by Jiangsu Bolong Yuhang New Material Technology Co., Ltd. (Zhenjiang, China). The specific parameters of the fabric prefabricated parts are shown in Table 1. The epoxy resin (P700-1M-A) and hardener (P700-1M-B) were mixed in a certain ratio and then degassed to remove air bubbles. Then, the matrix resin was injected into the 2.5D woven prefabricated part by the vacuum-assisted resin transfer molding (VARTM) technique, at 0.3 MPa, and cured at 80 °C for 8 h before being prepared for removal from the mold when cooled to 60 °C (shown in Figure 1c) [35,36,37]. By the wire-cut machinery, a total of 9 specimens with dimensions of 15 mm × 15 mm × 15 mm were cut out and tested in the radial, weft and thickness directions, with 3 repetitive tests in each direction (shown in Figure 1e). The fiber volume fraction (Vf) of the specimen, determined by the muffle combustion method, was 41.20%.

### 2.2. Mechanical Test

The size of the compression test sample was 15 mm × 13 mm × 15 mm. Specimen sizes are shown in Figure 2d. A total of 3 sets of specimens were cut for the compression tests, and a total of 9 samples were obtained. As shown in Figure 2, the samples were tested on the MTS Landmark 370 electro-hydraulic servo universal materials testing machine for the compression test in three directions. The compression experiment was conducted according to the standard GB/T 1448-2005 (test method for compression properties of fiber-reinforced plastics). The loading speed was 2 mm/min. The test was carried out under the conditions of one atmosphere pressure and a constant temperature of 25 °C.

XR-CT was used to observe the microscopic damage pattern of the compressed specimens. For XR-CT sample preparation, compression specimens measuring 15 mm × 13 mm × 15 mm were glued to the sample holder using an adhesive, and the beam emitted by the X-ray source in the machine passed through the compressed specimen and formed a projected image on the detector. This experiment was performed with a high-resolution XR-CT machine named ZEISS Xradia 610 Versa (shown in Figure 3).

## 3. Results

### 3.1. Compression Performance Analysis

The specimen was tested, and the raw data were processed to obtain the maximum load and compression strength of the specimen (shown in Table 2, Table 3 and Table 4). The compressive load-displacement curve for the material is shown in Figure 4 The compression strength $\sigma_{c}$ was calculated by the following formula [38]:(1)$$\sigma_{c}=\frac{P}{F}$$
where *P* was the maximum load of the compression test, N, and *F* was the cross-sectional area of the specimen, mm^2^.

Experimental data for compression in the warp, weft, and thickness directions are shown in Figure 4 and Table 2, Table 3 and Table 4. As can be seen from the load-displacement curves, the trend of compressive loading in the longitudinal, latitudinal, and thickness directions of the specimen was approximately the same in the initial stages of the test. The load to which the material was subjected tended to rise linearly with displacement. As the platen moved down, the material reached its yield point in both the warp and weft directions, and underwent plastic yielding, with material failure finally occurring. The material did not yield before failure in the thickness direction. When the material reached its maximum load, the material was rapidly unloaded, indicating that a large degree of damage had been done to the specimen. This was because shear damage occurred mainly in the thickness direction of the material. After the formation of the shear zone, the relative slip of material occurred under compression load and a sharp drop in load on the material. There were large differences in peak loads in different directions. The binder yarns were interwoven with the weft yarns at an angle in the thickness direction, to give the material better integrity. Therefore, the mechanical properties of the composite in the thickness and weft direction were significantly better than the mechanical properties in the warp direction.

### 3.2. Failure Mechanisms

In general, the fibers in the direction of loading contributed significantly to the mechanical properties of the material [39,40]. As can be seen from Figure 4, the carrying capacity of the compressed specimen, when loaded in the warp direction, was significantly less than the carrying capacity of the specimen in the weft and thickness directions. This was due to the flexion of the yarn in the warp direction. When the binder yarn was subjected to compressive loads, this resulted in a more easily altered structure in that direction. Figure 5 shows the typical deformation and damage pattern of the specimen in the warp direction under quasi-static compression testing. Figure 5a shows that when the platen was moved down, the material matrix was first subjected to compressive loads and cracks occurred in the resin matrix on the surface of the material. Subsequently, the matrix cracks extended along the interface between the fiber bundle and the matrix, the material appeared partially delaminated, and the binder warp yarns were squeezed and dispersed (Figure 5b). The material was subjected to compressive loads along the warp direction, and gradually reached yield, and then the material failed. As shown in Figure 5a, deformation and damage occurred at the edges of the material due to the kink effect in the position of the yarn curl. Cracking and delamination of the resin matrix, and localized macroscopic cracks, formed by fiber bundle dispersion, were the main mechanisms of failure in the warp specimen. The specimen was compressed along the weft direction as shown in Figure 6. During compression, the weft yarn in the straightened state had good carrying capacity, and contributed to the compressive strength and modulus of the material in the weft direction. The weft specimens showed matrix microcracks and damage at the fiber bundle/matrix interface under compressive loading (Figure 6b). The binder yarns, which were interwoven with the weft yarns at an angle in the thickness direction, joined the parallel weft yarns together, preventing damage at the fiber/matrix interface from spreading in all directions and making the material less susceptible to crushing. Therefore, the compression strength of the specimen in the weft direction was better than that in the warp direction. The weft yarns that were in a straightened state on the side of the material became spread out when subjected to compressive loads (Figure 6a). Fluctuations in the fiber bundle led to extensive fiber bundle/matrix interface damage and tow crash, but the degree of damage was relatively low.

As shown in Figure 7, the specimen was compressed along the thickness direction. Figure 7a shows a shear zone formed along with the specimen at an angle of 45° to the thickness direction, with damage dominating; the warp and weft yarns through which the sheared zone passes were all broken, and the resin around it was also almost broken, which was confirmed by the CT scan in Figure 7d. As shown in Figure 7c, the warp yarns along the shear zone were all broken. The main carrying section of the warp yarns was the inclined fiber bundle in the thickness direction. When the material was subjected to compressive loads, the warp yarns in the inclined direction stretched laterally, and the fiber bundles were subjected to tensile loads, then the compressive loads spread along the warp yarns to the interior of the material. The warp yarns were in contact with the horizontally arranged weft yarns, where the pressure load was concentrated. As the pressure built up, the resin/fiber interface was broken down, and resin debris was produced. When the fibers were stretched to their strength limit, the warp yarns broke. This was manifested macroscopically as a shear band at an angle of 45° along with the thickness. As shown in Figure 7b, with the appearance of the shear band, the resin near the shear band slipped relatively, and the straightened weft yarn was subjected to greater shear loads. At the same time, the bent part of the warp yarns and the straightened weft yarns were squeezed against one other, and the stress was concentrated in this area; the weft yarns were broken when the stress concentration in this area reached its failure limit.

## 4. Conclusions

In this study, 2.5D weave composite materials were developed by high-performance carbon fiber and epoxy resin using the VARTM process. XR-CT images were used to show the microstructure and failure morphologies of the 2.5D woven composites. The results showed that the mechanical response of the material to the compressive load was different in different directions. The maximum average load carrying capacity of the material in the thickness direction was 47.06 KN, while the maximum average load carrying capacities in the warp and weft directions were 12.90 and 23.93 kN. The compression properties in the thickness direction were superior to those in the weft and warp directions. As the state of the fibers located in the direction of loading had a great influence on the mechanical properties of the material, there was a great difference in the compressive strength of the material in different directions. The compression results showed that the compression strengths in the warp and weft directions were 65.18 and 105.13 Mpa respectively. This was because warp yarns in flexure caused the structure in that direction to change more easily, so that the compression strength was lowest in the warp direction. The presence of weft yarns in the straightened state, and the binding and stitching of warp yarns, increased the compression strength in the weft direction. The compression strength in the thickness direction was 239.92 MPa, which was higher than that in the warp and weft directions. This was because the warp and weft yarns, as a whole, shared the compression load. Combining the images showed that matrix cracking, yarn fracture, and crush as well as interfacial damage were the main modes of compression damage during compression of the material in the warp, weft, and thickness directions. This study can help to determine the mechanical properties and failure modes of 2.5D woven composites for end-use products, and will provide reference for subsequent related research. 

## Figures and Tables

**Figure 1 materials-15-03953-f001:**
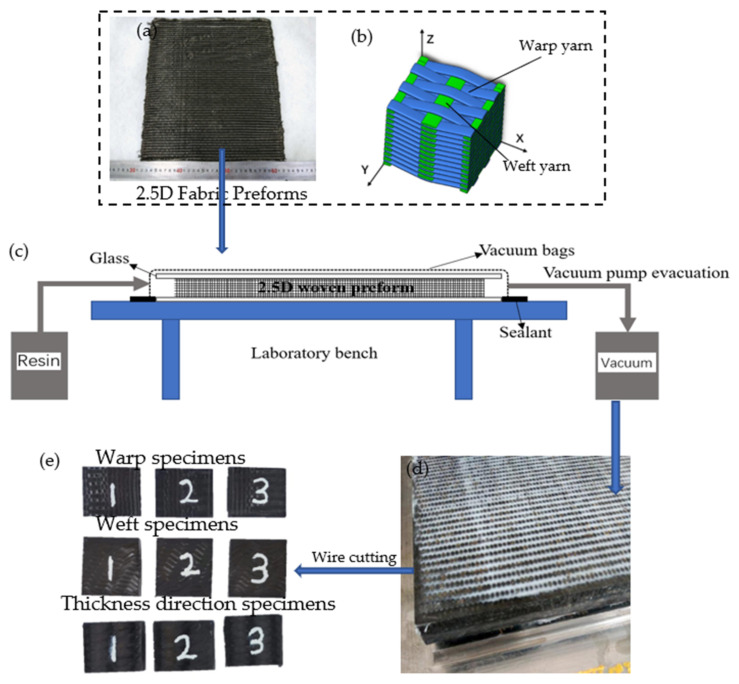
Preparation of 2.5D specimen (**a**) 2.5D woven prefabricated parts; (**b**) schematic diagram of the model; (**c**) schematic diagram of the VARTM; (**d**) 2.5D woven composite sample; (**e**) compression specimens.

**Figure 2 materials-15-03953-f002:**
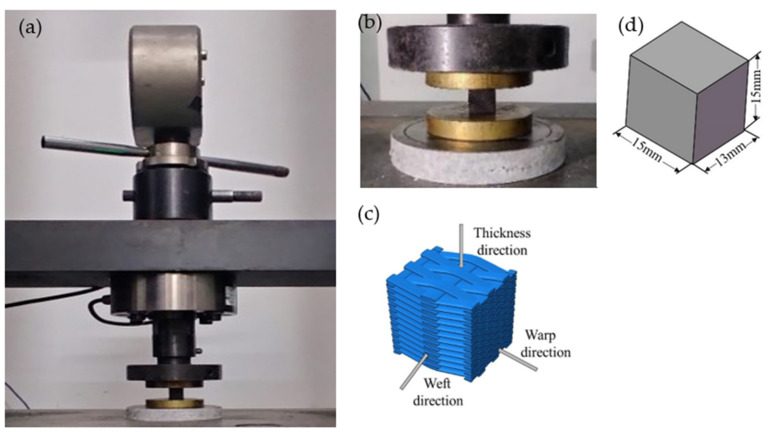
Compression test diagrams: (**a**) specimen under compression load; (**b**) experimental specimen closer view; (**c**) loading diagram; (**d**) specimen size.

**Figure 3 materials-15-03953-f003:**
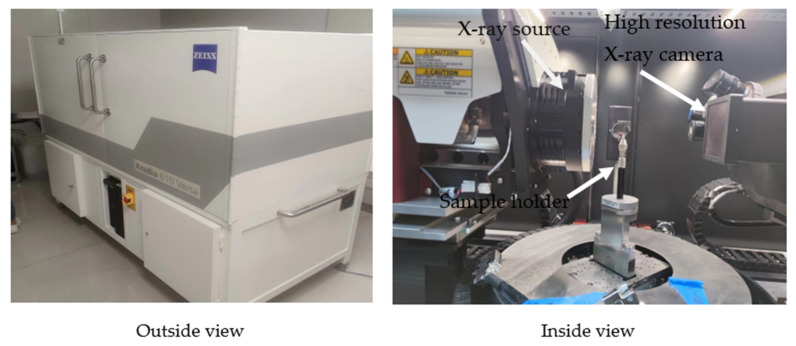
X-ray computed tomography machine.

**Figure 4 materials-15-03953-f004:**
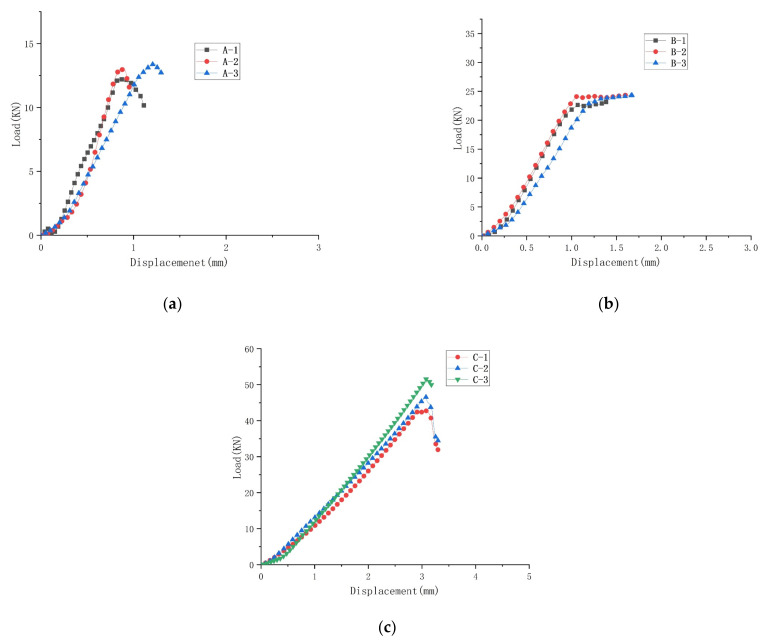
Quasi-static compression load-displacement curves for specimens in the (**a**) warp, (**b**) weft, and (**c**) thickness directions.

**Figure 5 materials-15-03953-f005:**
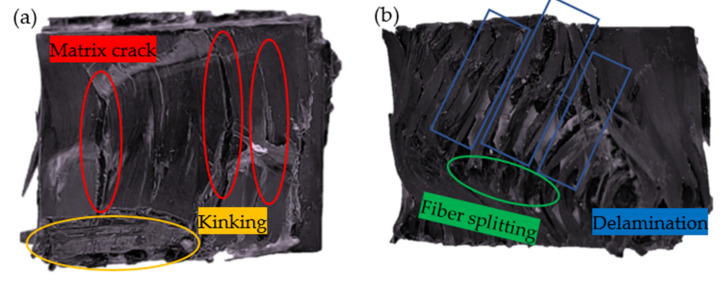
Damage pattern of quasi-static compression test in the warp direction: (**a**) top view; (**b**) side view.

**Figure 6 materials-15-03953-f006:**
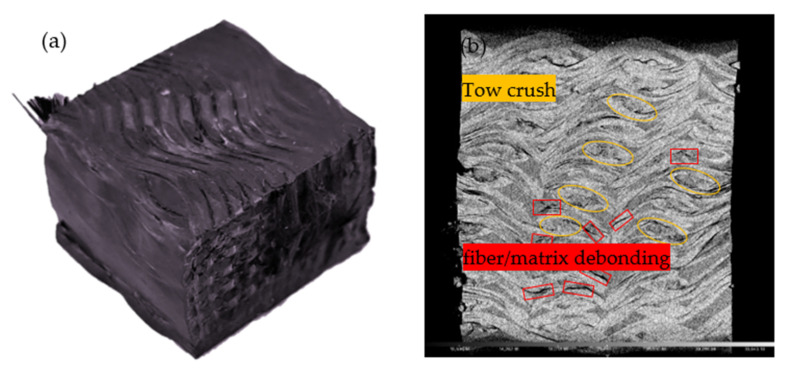
Damage pattern of quasi-static compression test in the weft direction: (**a**) macroscopic damage images; (**b**) internal microscopic images.

**Figure 7 materials-15-03953-f007:**
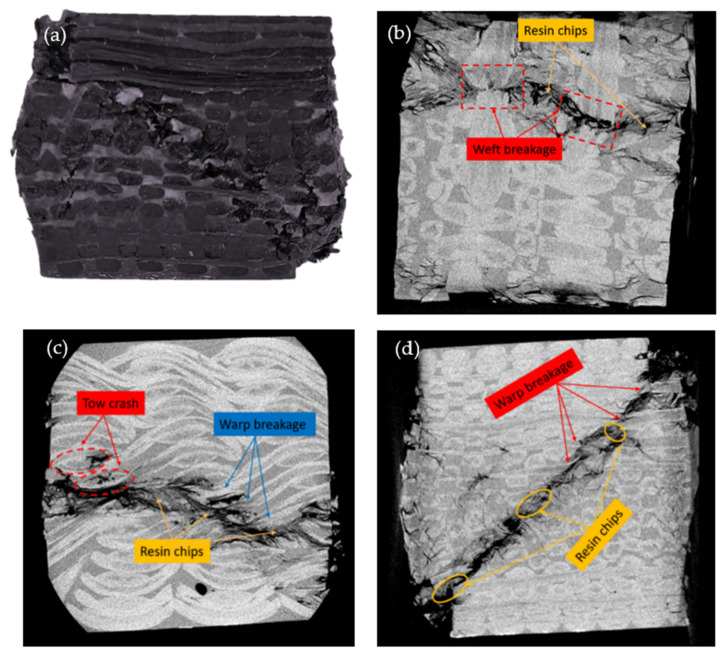
Damage pattern of quasi-static compression test in the thickness direction: (**a**) macroscopic damage images; (**b**) warp direction slicing; (**c**) weft direction slicing; (**d**) thickness direction slicing.

**Table 1 materials-15-03953-t001:** Prefabricated fabric parameters.

Specification Parameters	Value
Raw materials	T300-12K carbon fiber
Structure	2.5D angle-interlock
Specification	Warp: 12K × 1 Weft: 12K × 2
Size (mm)	330 × 300 × 15
Density (roots/cm)	Warp: 9 Weft: 2
Thickness (mm)	15

**Table 2 materials-15-03953-t002:** Quasi-static compression test data for warp specimens.

Specimens in Warp Direction	Length (mm)	Width (mm)	Height (mm)	Peak Load (KN)	Compression Strength (Mpa)
A-1	15.32	13.02	15.21	12.25	61.86
A-2	15.24	12.93	15.17	13.02	66.38
A-3	14.89	12.98	15.36	13.42	67.31
Average value	15.15	12.98	15.25	12.90	65.18
Standard deviation	0.19	0.04	0.08	0.49	2.38
Coefficient of Variation	1.25	0.31	0.52	3.80	3.65

**Table 3 materials-15-03953-t003:** Quasi-static compression test data for weft specimens.

Specimens in Weft Direction	Length (mm)	Width (mm)	Height (mm)	Peak Load (KN)	Compression Strength (Mpa)
B-1	15.21	13.01	14.94	23.16	101.92
B-2	15.14	12.95	15.12	24.34	106.33
B-3	14.98	13.12	15.14	24.30	107.14
Average value	15.11	13.03	15.07	23.93	105.13
Standard deviation	0.10	0.07	0.09	0.55	2.29
Coefficient of Variation	0.66	0.54	0.60	2.30	2.18

**Table 4 materials-15-03953-t004:** Quasi-static compression test data for thickness specimens.

Specimens in Weft Direction	Length (mm)	Width (mm)	Height (mm)	Peak Load (KN)	Compression Strength (Mpa)
C-1	15.04	13.08	15.11	42.75	217.31
C-2	15.12	12.96	15.13	46.56	237.61
C-3	14.96	13.09	15.02	51.86	264.83
Average value	15.04	13.04	15.09	47.06	239.92
Standard deviation	0.07	0.06	0.05	3.74	19.47
Coefficient of Variation	0.47	0.46	0.33	7.95	8.12

## Data Availability

Data sharing not applicable.

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
