# Peer review of "Quasi-Static Compression Response of Carbon Fiber Reinforced 2.5D Woven Composites at Different Loading Directions"

_materials, 2022, doi:10.3390/ma15113953_

Round 1
Reviewer 1 Report
Dear authors, thank You for this interesting paper. I think that the presented material should also take into account a larger literature review of works from the Chemnitz University as there are a lot of papers published in this specific topic. As for the material it is mainly concerned with the static tests, which is an improtant issue for many readers occupied with these types of materials. In my opinion the information provided by the authors should be expanded in terms of statistics as we didn't get a clear information about the error of the experiment. As for some minor issues e.g.:
Page 1 paragraph 37: ....Duan et al..... missing citation
Page 2 paragraph 52: ...Naik et al. .... missing citation
the same case applies for other missed citations.
The name of the last part of the paper should be renamed or new conclussions should be added. Conclusions part is more a discussion or observation part in this paper. In the current state the conclusions are not supported by experimental results.
Reviewer 2 Report
Comments
Modifications are sought to respond to the comments listed below.
- The samples considered for the flexural test should be listed in the abstract for a better understanding for the readers.
- The percentage increment in the mechanical properties compared to the reference samples can be added in the abstract.
- The current state-of-the-art considering 2.5D angle-interlock woven composites is weak. This may be because this was poorly considered in the literature review or little information exists.
- The clarity in this area will aid the reader in better appreciating the motivation for this work.
- The mechanism behind improving the through the thickness properties of composite laminates considered in this paper and other strategies to improve the energy absorption characteristics can be explained further in the introduction. The authors can also add other relevant literature.
https://doi.org/10.1016/j.compstruct.2020.113519, https://doi.org/10.1177/0021998319856426, https://doi.org/10.1016/j.polymertesting.2017.05.014, https://doi.org/10.1002/pc.24884, https://doi.org/10.1016/j.compstruct.2019.111007, https://doi.org/10.1016/j.compstruct.2015.08.022
- The preparation of the 2.5D woven fabric should be provided in detail with a schematic diagram. Further, the processing parameters employed to fabricate the composite vis RTM should be explained.
- A photographic or schematic image showing the three different loading directions wrt the samples should be added in Section 2.
- The CT scan procedure can also be added in Section 2.
- “As can be seen from the load-displacement curve, the trend of compression loading in the longitudinal, latitudinal, and thickness directions of the specimen is approximately the same.” The compression curves show different behavior after yielding in Figure 5. Please check this.
- From Table 2, why is the compression strength in the thickness direction higher than in the weft direction?
- The legends and labels given in the figures are not visible. Please modify them.
- The results for normal 2D samples are missing.
- The explanation given in section 3.2 can be discussed together comparatively (Instead of in three separate paragraphs). Now it looks more like a report than a research article. If possible, this section can also be combined with section 3.1.

Reviewer 3 Report
This manuscript is aimed to prepare a 2.5D woven composites structure. The manuscript didn’t write well and doesn’t enough novelty. It was prepared one sample and examined its properties on different direction which isn’t sufficient. More over there is unclear parts as a following comments;
The abbreviation should to introduce at the first time presenting such as RTM
Page 3, line 90; using “under set pressure” is unclear
Page 3, line 99; it is reported that “Specimen sizes are shown in Figure 4.” While didn’t present.
Round 2
Reviewer 1 Report
Dear authors thank You for the revision. Once again the conclusion part of the paper should be remodelled as some of the statements are more of an observation than supported by results. What does it mean: "better"? These statements should be supported by numerical values. As for the references a lot of new references have been added which really extends the literature review. I recommend to accept after minor revision.
Reviewer 3 Report
The manuscript is acceptable at the current format.
Author Response
Dear Reviewer,
On behalf of my coauthors, I would like to thank you for reviewing our manuscript and recommendations for acceptance.